# Rapid Removal of Azophloxine via Catalytic Degradation by a Novel Heterogeneous Catalyst under Visible Light

**Di Wu, Kequan Xia, Chengzhu Fang, Xuegang Chen * and Ying Ye**

Ocean College, Zhejiang University, Zhoushan 316021, China; w655460@163.com (D.W.);
KequanXia@163.com (K.X.); fangczwork@163.com (C.F.); gsyeying@zju.edu.cn (Y.Y.)
* Correspondence: chenxg83@zju.edu.cn; Tel.: +86-580-209-2326; Fax: +86-580-209-2891

**Abstract:** Azo dyes are the most widely used synthetic dyes in the printing and dyeing process. However, the discharge of untreated azo dyes poses a potential threat to aqueous ecosystems and human health. Herein, we fabricated a novel heterogeneous catalyst: activated-carbon-fiber-supported ferric alginate (FeAlg-ACF). Together with peroxymonosulfate (PMS) and visible light, this photocatalytic oxidation system was used to remove an azo dye—azophloxine. The results indicated that the proposed catalytic oxidation system can remove 100% of azophloxine within 24 min, while under the same system, the removal rates were only 92% and 84% when ferric alginate was replaced with ferric citrate and ferric oxalate, respectively, which showed the superiority of FeAlg-ACF. The degradation of azophloxine is achieved by the active radicals ($SO_4^{•−}$ and $•OH$) released from PMS and persistent free radicals from activated carbon fiber. Moreover, due to ferric alginate's highly intrinsic photosensity, visible radiation can further enhance the ligand-to-metal charge transfer (LMCT) processes. After 24 min of treatment, the total organic carbon of the azophloxine solution (50 μmol/L) decreased from 1.82 mg/L to 79.3 μg/L and the concentration of nitrate ions increased from 0.3 mg/L to 8.6 mg/L. That is, up to 93.5% of azophloxine molecules were completely degraded into inorganic compounds. Consequently, potential secondary contamination by intermediate organic products during catalytic degradation was prohibited. The azophloxine removal ratio was kept almost constant after seven cycles, indicating the recyclability and longevity of this system. Furthermore, the azophloxine removal was still promising at high concentrations of $Cl^−$, $HCO_3^−$, and $CO_3^{2−}$. Therefore, our proposed system is potentially effective at removing dye pollutants from seawater. It provides a feasible method for the development of efficient and environmentally friendly PMS activation technology combined with FeAlg-ACF, which has significant academic and application value.

**Keywords:** peroxymonosulfate; ferric alginate; activated carbon fiber; visible radiation; heterogeneous photocatalysis

## 1. Introduction

Azo dyes are widely used in the printing and dyeing industries. However, about 10%–15% of residual azo dyes are discharged into the environment without treatment, which causes severe environmental pollution and poses hazards for aqueous organisms and human health [1–4]. Therefore, it is of great practical significance to develop efficient and environmentally friendly treatment methods for dye wastewater. Among many water pollution treatment methods, Fenton-like oxidation technology, as a typical advanced oxidation technology, has been widely favored by researchers due to its advantages of high efficiency, environmental friendliness, and circularity [5–9]. The amount of

reactive oxygen species (ROS) is an important factor for the catalytic efficiency of the photocatalytic system, especially in relatively mild conditions [10–13]. Peroxymonosulfate (PMS) is a widely used material to provide ROS to remove a variety of refractory organic pollutants in environmental remediation [14,15]. PMS can be activated to produce $\bullet OH$ and $SO_4^{\bullet-}$ by transition metal ions with advantages of simple reaction conditions, rapid reaction rate, and low cost [16–18].

Studies have shown that Fe(III) easily forms complexes with carboxyl groups [19–22]. These complexes are often photoactive and can be decomposed under light to produce a variety of active species (Equations (1)–(7)):

$$RCOO\text{-}Fe(III) + hv \rightarrow Fe(II) + RCOO \tag{1}$$

$$RCOO\bullet \rightarrow R\bullet + CO_2 \tag{2}$$

$$R\bullet + Fe(II) \rightarrow Fe(III) + products \tag{3}$$

$$R\bullet + O_2 \rightarrow O_2^-\bullet + products \tag{4}$$

$$H^+ + O_2^-\bullet \rightarrow HO_2\bullet \tag{5}$$

$$HO_2\bullet + HO_2\bullet \rightarrow H_2O_2 + O_2 \tag{6}$$

$$HO_2\bullet + H^+ + Fe(II) \rightarrow H_2O_2 + Fe(III). \tag{7}$$

In analogy to other Fe(III)–carboxylate complexes, such as ferric citrate and ferric oxalate [23–25], ferric alginate (Fe-Alg) could initiate the generation of a series of active oxygen species and Fe(II)/Fe(III) ions by the ligand-to-metal charge transfer (LMCT) under light irradiation [26]. However, ferric citrate and ferric oxalate are greatly affected by the pH value. When the pH value is greater than 4, both the ferric citrate and ferric oxalate complexes are converted into $Fe_2O_3\bullet nH_2O$ amorphous precipitation, leading to the decrease or disappearance of the optical activity of the system [27].

As an iron-based photocatalyst, the Fe-Alg complex is made by cross-linking the multivalent Fe(III) with natural, nonhazardous, and edible polysaccharide alginate [28]. There are a large number of carboxyl groups in the molecular structure of alginate. Studies have shown that some carboxyl acids can produce active reducing free radicals by self-photolysis, and there are also a large number of hydroxyl groups. Thanks to their unique gelling properties, alginates can react with metal ions to form stable organic–inorganic hybrid composite materials, which may be promising for applications in the environmental purification and remedy areas [29–33].

Although iron alginate gel beads that can increase the pH range of Fenton-like catalysts have been studied in recent years [34], there are still some unresolved difficulties, such as tedious preparation process, non-reusability, and hydrolysis under a strong base [35–37]. Therefore, it is necessary to construct a heterogeneous catalytic system with Fe-Alg as the active center to accelerate the process of metal–ligand charge transfer and improve the stability of materials, so as to improve the degradation efficiency of dye wastewater.

A suitable carrier is also essential for a catalytic system. The most commonly used heterogeneous catalytic carriers are zeolite, resin, clay, molecular sieve, activated carbon, etc. [38–41]. Among them, activated carbon has attracted extensive attention due to its large specific surface area and strong adsorption performance. However, activated carbon is generally powdered or granular, and there are still problems such as separation and regeneration difficulties between catalyst and reaction medium. Compared with powdered or granular activated carbon, activated carbon fiber (ACF) has

more excellent properties [42]. Due to its microporous structure, ACF usually shows a specific surface area of up to 3000 m$^2$/g with an average value of 1000–1500 m$^2$/g [43–45]. The outer surface area of ACF is two orders of magnitude higher than the internal surface area. Therefore, ACF could provide high adsorption efficiency for the photocatalytic system. The adsorption ability of ACF is stronger than that of activated carbon for some macromolecules or particles [46,47]. Furthermore, the surface of ACF contains abundant oxygen-bearing groups, and its surface structure can be modified to further improve its adsorption performance [48,49]. On the other hand, ACF could provide large amounts of persistent free radicals (PFRs) for Fe, which helps to further accelerate the LMCT process and reduce Fe(III) to the Fe(Il) state [15,50]. Therefore, when transition metals such as Fe, Cu, Co, and Ni are loaded on the surface of ACF, through ACF's inherent PFRs starting the electron transfer in catalytic oxidation and continuing to provide electrons, those transition metals can accelerate high transition metal reduction reaction with visible light photocatalytic synergy, activate the PMS to produce ROS, and thus degrade organic dyes.

In this work, we report a novel photocatalytic oxidation system (FeAlg-ACF/PMS/visible light) for the first time. The ACF serves as the carrier, and PMS provides oxidants; Fe-Alg was loaded on the surface of ACF via a simple impregnation method to create a Fenton-like catalyst, FeAlg-ACF. We assessed the removal of azo dyes by this novel system and investigated the optimum treatment conditions. In addition, we propose the potential degradation mechanism of azo dyes by FeAlg-ACF/PMS under visible light.

## 2. Results and Discussions

### 2.1. Characteristics of the As-Prepared Catalyst

The surface topography of ACF and FeAlg-ACF was evaluated by SEM. Figure 1a,c respectively shows the surface morphologies of ACF and FeAlg-ACF at 5000 times magnification. The iron alginate was densely and uniformly distributed on the surface of the ACF. In addition, the iron content of the precipitated solution after the formation of the heterogeneous catalyst FeAlg-ACF was determined to be only 0.026% by XRF. This indicated that iron had been successfully loaded onto the ACF. The microscopy results suggested that Fe-Alg particles were densely and uniformly cross-linked and distributed on the surface of ACF. At high magnification, as shown in Figure 1b,d, the loaded Fe-Alg particles showed rod-like shapes with a width of about 40 nm and a length of 60 nm.

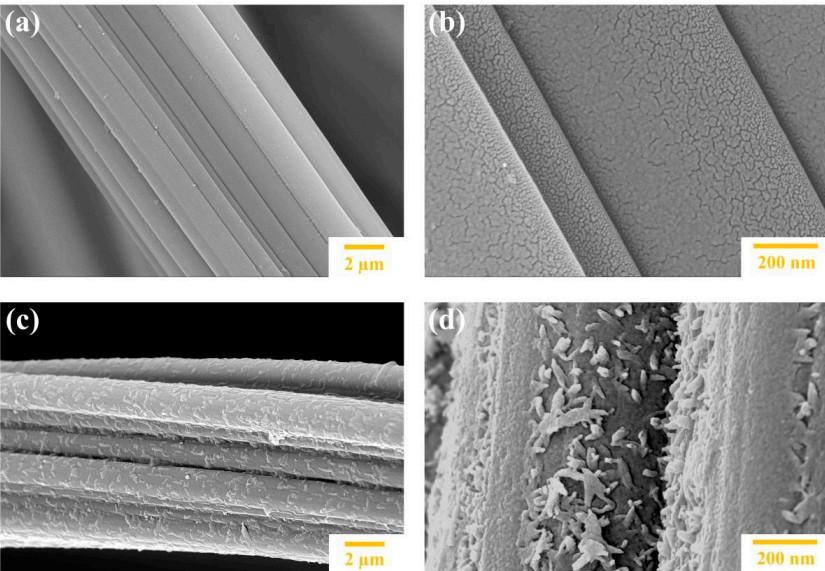

**Figure 1.** SEM images of activated carbon fiber (ACF) at 5000× (**a**) and 45,000× (**b**) magnification, as well as the FeAlg-ACF catalyst at 5000× (**c**) and 45,000× (**d**) magnification.

### 2.2. Dye Removal by Fe-Alg Catalyst

The removal of azophloxine by different systems was investigated in the dark and in visible light. As shown in Figure 2A, ACF and FeAlg-ACF removed 8% and 10% of azophloxine, respectively, after 24 min in dark conditions, suggesting that the adsorption of azophloxine by ACF and FeAlg-ACF was quite limited. The adsorption equilibrium was achieved after 12 min. By contrast, when PMS was introduced, about 47% of the azophloxine was removed by the Fe-Alg/PMS system. This indicated that PMS can significantly improve the removal of dyes because of its activation by Fe-Alg to generate $SO_4^{\bullet-}$ efficiently. Surprisingly, when the FeAlg-ACF/PMS system was exposed to visible radiation, almost all the azophloxine was rapidly removed from the solution within 24 min. Correspondingly, the characteristic absorption peak of azophloxine at 531 nm gradually decreased with time and totally disappeared after 24 min (Figure 2C). This result suggested that azophloxine was degraded rapidly by the FeAlg-ACF/PMS system under visible light irradiation, testified to by the decreasing total organic carbon content from 1.82 mg/L to 79 μg/L. In addition, the nitrate ($NO_3^-$) concentration increased from 0.3 mg/L before catalytic degradation to 8.6 mg/L after 24 min of treatment. According to the molecular structure of azophloxine, complete degradation of this dye with an initial concentration of 50 μmol/L would generate a nitrate content of 8.6 mg/L in the resulting solution. Consequently, about 93.5% of the azophloxine molecules were completely degraded into inorganic compounds after treatment with the FeAlg-ACF/PMS system, which paves the way for developing high-efficiency processes to be subsequently used in environmental catalysis.

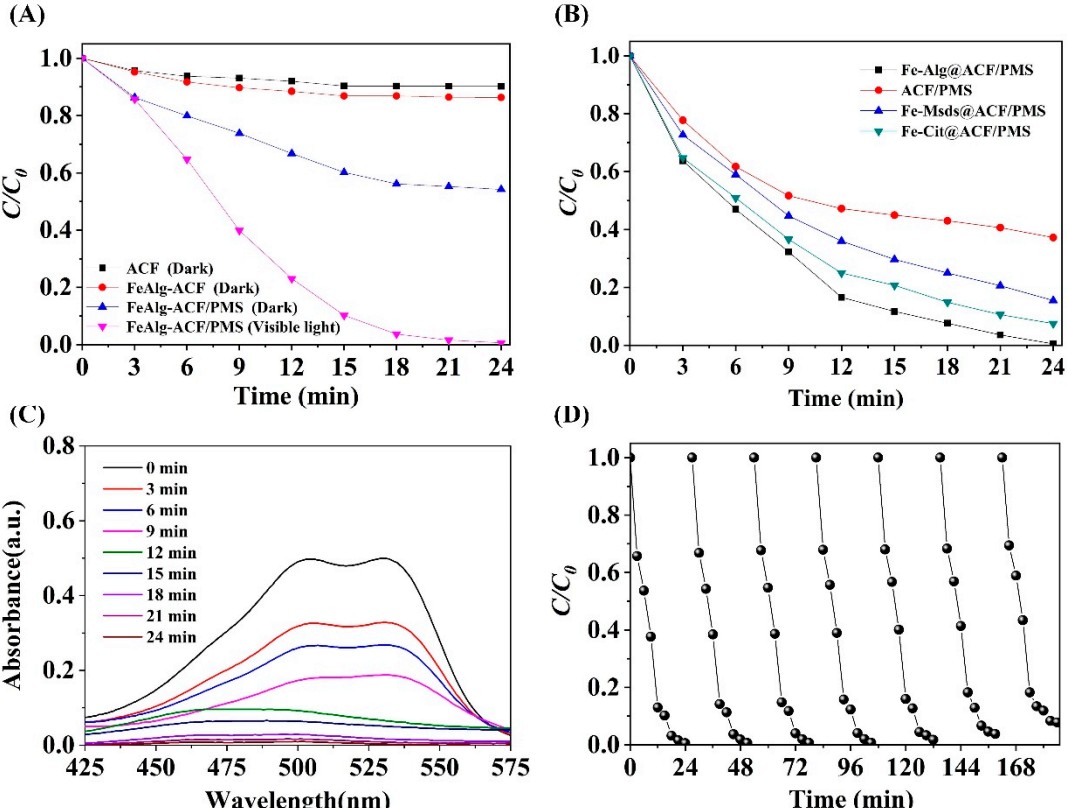

**Figure 2.** (**A**) Time-dependent profiles of azophloxine decolorization with and without light irradiation; (**B**) Decolorization of azophloxine by different systems as a function of time; (**C**) UV–vis spectra of azophloxine solutions after removal by the FeAlg-ACF/peroxymonosulfate (PMS) system; (**D**) Azophloxine removal by FeAlg-ACF/PMS after a certain number of cycles. Reaction conditions: [FeAlg-ACF] = 2 g/L, [PMS] = 5 mmol/L, [azophloxine] = 50 μmol/L, initial pH = 4.0, T = 25 °C.

The effects of different systems on azophloxine removal are compared in Figure 2B. Comparative experiments with ferric alginate (Fe-Alg), ferric citrate (Fe-Cit), and ferric oxalate (Fe-Msds) using the same carrier and oxidant were tested. The results indicated that the proposed catalytic oxidation system can remove 100% of azophloxine within 24 min, while under the same system, the removal rates were only 92% and 84% when Fe-Alg was replaced with Fe-Cit and Fe-Msds, respectively. This is related to the photosensitivity of Fe-Alg and its unique complexation structure. When ACF reacted with the oxidant PMS alone, the removal rate was 60% after 24 min. This phenomenon is attributed to the fact that ACF is a porous carbon material which can adsorb 8% azophloxine. On the other hand, ACF will continuously provide PFRs as electronic storage to participate in the LMCT process, and under light irradiation, PMS is activated by electron transfer to produce small amounts of $SO_4^{\bullet-}$ and $\bullet OH$ to degrade dye.

Our prepared FeAlg-ACF/PMS system could sustain several cycles. As shown in Figure 2D, the azophloxine removal efficiency after 24 min remained almost constant after seven cycles, decreasing only slightly from 100% to 93%. This is ascribed to the high structural stability, repeatability, and recycling performance of the FeAlg-ACF photocatalyst. Meanwhile, our chosen oxidant PMS is stable during dye treatment, and the treatment process is in line with safety and environmental protection requirements. Furthermore, we also assessed the removal efficiency of our catalytic system for acid orange 7 and methylene blue. The achieved removal ratios of acid orange 7 and methylene blue were 98.0% and 97.4% by FeAlg-ACF/PMS under visible light irradiation after 24 min. Our system therefore shows great potential for the removal of reactive dyes, acid dyes, and other typical dyes, and shows promise for applications in dye wastewater treatment and environmental restoration.

## 2.3. Effect of Reaction Conditions

### 2.3.1. Effect of PMS Dosage

PMS produces ROS in the reaction system and provides oxidants for the catalytic degradation of organic pollutants by a catalyst. The removal ratio of azophloxine in our catalytic system varied with the initial PMS concentration, as shown in Figure 3A. After 24 min of treatment, the removal ratio was enhanced from 74% to 100% as the PMS dosage was increased from 0 to 5 mmol/L. At the PMS concentration of 5 mmol/L, azophloxine was completely degraded within 9 min when the initial pH was 4. The enhanced removal of azophloxine by increasing PMS concentration is attributed to the increased generation of ROS which are conducive to the catalytic reaction. In consideration of environmental protection, economy, and practicability, the initial content of PMS was set as 5 mmol/L in the following experiments. Besides this, we calculated the utilization efficiency of PMS using the oxidant consumption index ($X$): the number of azophloxine removed per mole number of oxidant consumed. A lower $X$ value indicates a higher utilization efficiency of oxidant [51–53]. We calculated that our catalytic system ($X = 31.4$) showed excellent PMS utilization compared with the Fe-Alg/PMS system ($X = 43.7$) and the FeAlg-ACF/PMS system ($X = 57.1$) without visible light radiation.

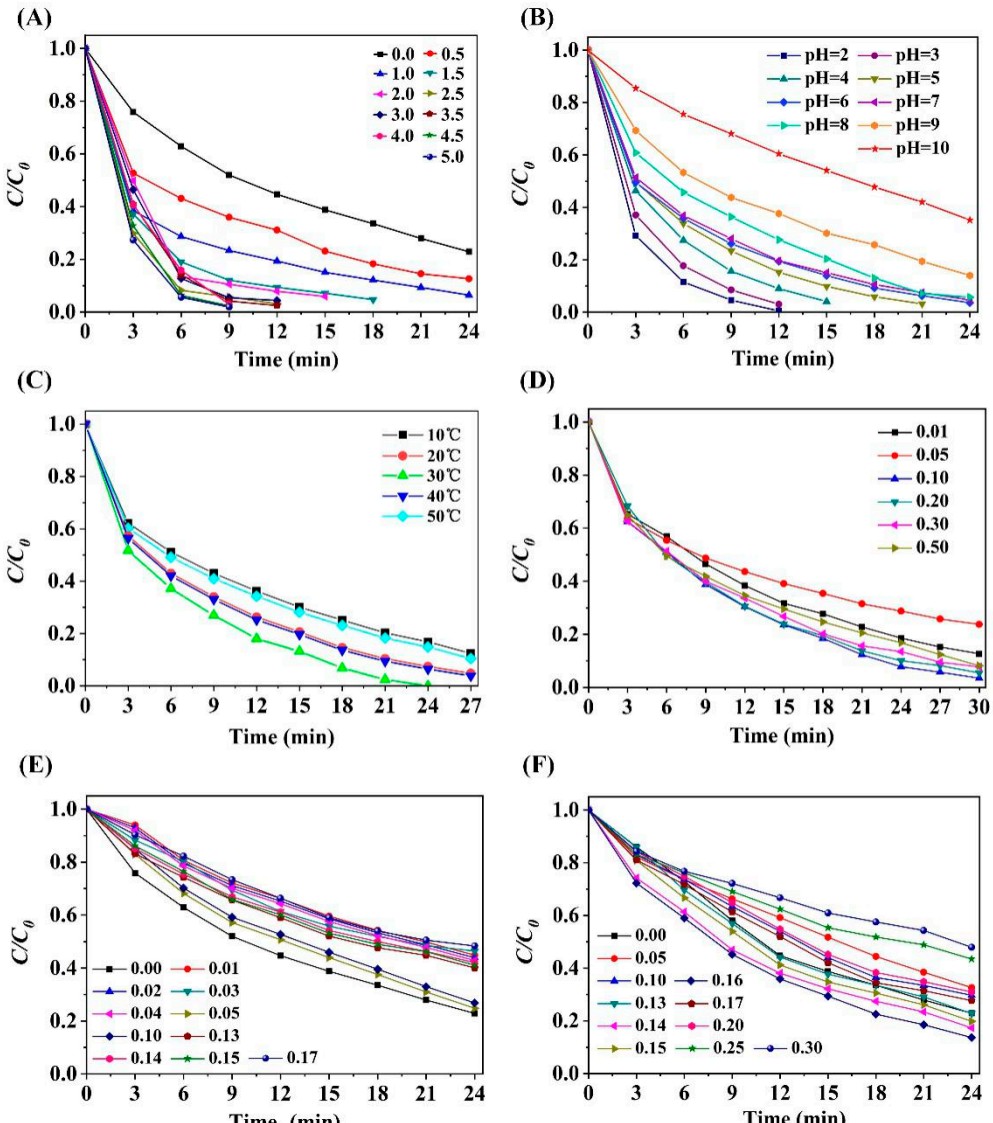

**Figure 3.** Time-dependent profiles of azophloxine removal under in different conditions: (**A**) Effect of PMS dosage. Reaction conditions: T = 25 °C, pH = 4; (**B**) Effect of initial pH. Reaction conditions: [PMS] = 5 mmol/L, T = 25 °C; (**C**) Effect of reaction temperature; (**D**) Effect of initial NaCl concentration; Concentration unit: mmol/L; (**E**) Effect of initial $Na_2CO_3$ concentration. Concentration unit: mmol/L; (**F**) Effect of initial $NaHCO_3$ concentration. Concentration unit: mmol/L; (**C–F**) Reaction conditions: [PMS] = 5 mmol/L, T = 25 °C, pH = 6. Other chemical reagents used: [FeAlg-ACF] = 2 g/L, [azophloxine] = 50 μmol/L.

### 2.3.2. Effect of Initial pH Value

The influence of the initial pH value on azophloxine removal is illustrated in Figure 3B. More than 95% of azophloxine was removed by the FeAlg-ACF/PMS system at pH values of 2.0–8.0 within 24 min when other parameters were kept constant. $S_2O_8^{2-}$ ions that are produced by PMS after activation can react with water or $OH^-$ to produce •OH. The $SO_4^{•-}$ and •OH mainly come from PMS photolysis, $SO_4^{•-}$ decay, and transformation of $SO_4^{•-}$ to •OH. When the pH is greater than 7, the ROS in the system are dominated by •OH. When the pH value is 7, $SO_4^{•-}$ is equally as involved in oxidation as •OH (Equations (8) and (9)).

$$SO_4^{•-} + H_2O \rightarrow SO_4^{2-} + •OH + H^+ \tag{8}$$

$$SO_4{}^{\bullet-} + OH^- \rightarrow SO_4{}^{2-} + \bullet OH \tag{9}$$

Compared with the typical Fenton system, our system shows excellent degradation performance under weak alkaline conditions. This is because as the pH value increases, the carboxyl dissociation on alginate is in a state of negative charge, and the pores number of the gel structure increase under the action of electrostatic repulsion, so the ability to absorb dye molecules is enhanced. The removal ratio of azophloxine generally decreases with increasing pH value (Figure 3B). When the initial pH value was 2.0, the time required to completely remove azophloxine was only 12 min, which was quite superior when compared with other photocatalytic Fenton systems [54,55]. At an initial pH value of 10.0, however, the azophloxine removal ratio significantly declined to only 65% after 24 min, but there was still a downward trend. This result also indicated that our FeAlg-ACF/PMS could effectively remove azophloxine in a wide pH range of 2–10. By contrast, traditional homogeneous or heterogeneous Fenton systems only work in narrow pH ranges (usually 2–4) [56,57].

### 2.3.3. Effect of Treatment Temperature

Temperature is an important factor affecting the process of activating PMS to degrade organic matter. Increasing the temperature can not only promote molecular thermal movement which improves the reaction rate of wastewater treatment but can also make the reaction system more easily overcome the reaction activation energy. As shown in Figure 3C, when the reaction temperature was lower than 30 °C, the removal of azophloxine increased with increasing temperature. When the temperature was higher than 30 °C, on the contrary, the photocatalytic degradation of azophloxine generally declined with increasing temperature. When the reaction temperature reached 50 °C, the azophloxine removal ratio was 90% after processing for 27 min. Because room temperature was close to the optimal reaction temperature (30 °C), the reaction temperature was set at 25 °C in the following experiments to facilitate the operation of experiments.

### 2.3.4. Impact of Ion Strength

Chloride can promote the transfer of dye from the aqueous phase to the fibrous phase and is thus widely used in the printing and dyeing industry. Besides this, there is a mass of chloride, carbanion, and bicarbonate ions in the seawater system [58]. Therefore, it is necessary to study the effect of chloride, carbanion, and bicarbonate ions on the oxidation activity of the catalytic system.

Figure 3D shows the change in the azophloxine removal rate in our catalytic system under different concentrations of NaCl. As shown in Figure 3D, when the NaCl concentration increased from 0 to 0.05 mmol/L, the removal rate of azophloxine in our catalytic system decreased, which could be attributed to the scavenging effect of •OH in NaCl. In our catalytic system, when the NaCl concentration increased from 0.05 mmol/L to 0.1 mmol/L, the azophloxine removal rate increased. When the chloride concentration was 0.1 mmol/L, the azophloxine removal rate was the highest, reaching 98% removal rate within 30 min. Chloride ions promoted pollutant removal. However, when the NaCl concentration exceeded 0.1 mmol/L, with further increase of the NaCl concentration, the azophloxine removal rate decreased; this showed that a high concentration of chloride inhibited azophloxine degradation.

The effect of widespread carbanion ($CO_3{}^{2-}$) ions in seawater on azophloxine degradation is illustrated in Figure 3E: when the concentration of $CO_3{}^{2-}$ increased, the removal rate of azophloxine became faster, which demonstrated that the introduction of $CO_3{}^{2-}$ promoted the removal of azophloxine. The introduction of $HCO_3{}^-$ had a slight inhibitory effect on the removal of azophloxine, and with increasing $HCO_3{}^-$ concentration, an overall effect appeared of first promoting and then inhibiting, as presented in Figure 3F. Therefore, we speculate that a low concentration of $HCO_3{}^-$ is involved in the pathway to promote the generation of •OH and $SO_4{}^{\bullet-}$, thus promoting the removal of azophloxine, while an excessive concentration of $HCO_3{}^-$ could capture ROS generated by the reaction to restrain the removal of azophloxine.

## 2.4. Comparison between the FeCit-ACFs/PMS/Visible Light and FeAlg-ACFs/PMS/Visible Light Systems

The Fe(III)–carboxylate complex, FeCit-ACFs/PMS/visible light, was tested for degradation under the same conditions. This system is able to degrade azophloxine within 33 min, while our catalytic oxidation system, FeAlg-ACFs/PMS/visible light, in the same conditions can degrade azophloxine within 21 min, and the reusability is better.

To further assess the difference in catalytic oxidation activity of these two systems, we adopted a general pseudo-first-order kinetic model ($\ln(C_t/C_0)/k_{obs}t$). As shown in Figure 4, the $k_{obs}$ for FeAlg-ACFs/PMS/visible light was calculated to be 0.19694 min$^{-1}$, which was almost twofold higher than that for FeCit-ACFs/PMS/visible light (0.1098 min$^{-1}$).

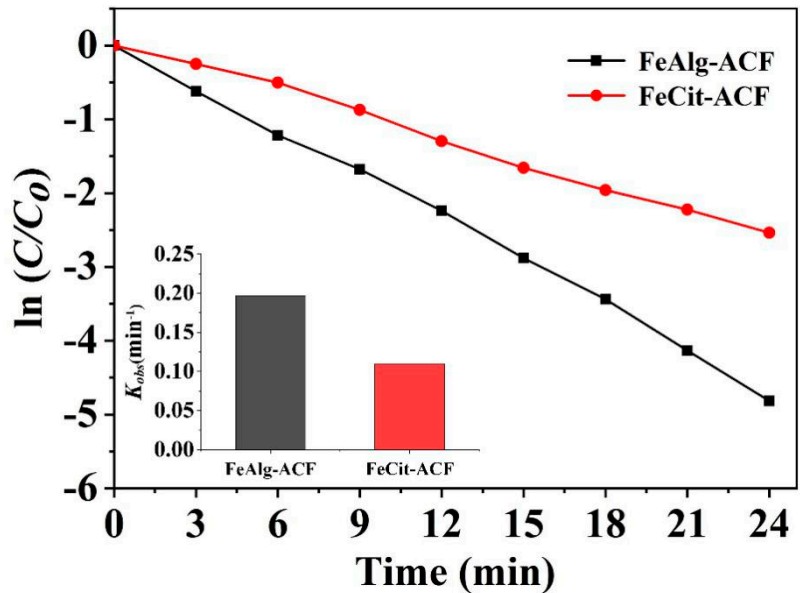

**Figure 4.** Time profiles of azophloxine decolorization in FeAlg-ACFs/PMS/visible light and FeCit-ACFs/PMS/visible light; the inset shows the $k_{obs}$ values of the corresponding systems. Reaction conditions: [PMS] = 5 mmol/L, T = 25 °C, pH = 6. Other chemical reagents used: [FeAlg-ACF] = 2 g/L, [FeCit-ACF] = 2 g/L, [azophloxine] = 50 μmol/L.

## 2.5. Reaction Kinetics of Catalytic Oxidation by the FeAlg-ACF/PMS System

Two models—pseudo-first-order and pseudo-second-order—were used to investigate the degradation kinetics of azophloxine degradation by the FeAlg-ACFs/PMS system. The pseudo-first-order model was first described by Lagergren (1898) and can be generally expressed as follows:

$$\log (q_e - q_t) = \log q_e - k_1 \, t/2.303 \tag{10}$$

where $q_t$ is the amount of azophloxine removed at time t (min) and $k_1$ is the equilibrium rate constant (min$^{-1}$). The values of $\log(q_e - q_t)$ are linearly related with t, while $k_1$ and $q_e$ can be determined from the slope and intercept of the plot of $\log(q_e - q_t)$ versus t, respectively.

A pseudo-second-order model, known as Ho's pseudo-second-order model, was also applied to analyze the degradation kinetics of azophloxine by using the equation below:

$$t/q_t = 1/(k_2 \, q_e^2) + t/q_e \tag{11}$$

where $k_2$ is the rate constant of second-order degradation (g mg$^{-1}$ min$^{-1}$) and $k_2 q_e^2$ is the initial degradation rate constant (h). The plot of $t/q_t$ versus t should exhibit a linear relationship, and $q_e$ and $k_2$ can then be determined from the slope and intercept of the plot, respectively.

These models were tested against the degradation of azophloxine and the best model was selected depending on the linear regression correlation coefficient, $r^2$. Table 1 shows the parameters of these degradation kinetic models. The pseudo-first-order kinetic model was $-\ln(C_t/C_0) = -0.19694t - 0.04125$, and the $R^2$ value was 0.99524, while the pseudo-second-order kinetic model was $1/C_t = -0.09478t - 21.84025$, and the $R^2$ value was 0.98076. We can conclude that the pseudo-second-order model is not suitable to model the degradation of azophloxine by the FeAlg-ACFs/PMS system because of its relatively low $r^2$ value. The experimental data fit well with the pseudo-first-order model with $r^2 > 0.99$.

**Table 1.** Kinetic constants of azophloxine degradation by the FeAlg-ACFs/PMS/visible light system.

| Dye | Dynamic Equation Series | The Fitted Reaction Rate Equation | $k_{obs}$ | Adj. R-Square |
|---|---|---|---|---|
| azophloxine | pseudo-first-order kinetic model | $-\ln(C_t/C_0) = -0.19694t - 0.04125$ | 0.19694 | 0.99524 |
| | pseudo-second-order kinetic model | $1/C_t = -0.09478t - 21.84025$ | 0.09478 | 0.98076 |

To better analyze the degradation rate of azophloxine, in-depth theoretical study of the degradation of the target pollutant azophloxine by the reaction system was conducted. Pseudo-first-order dynamics fitting and pseudo-second-order dynamics fitting were performed on the initial 24 min of azophloxine degradation. The fitting results are presented in Figure 5, and the fitting kinetic constants are presented in Table 1.

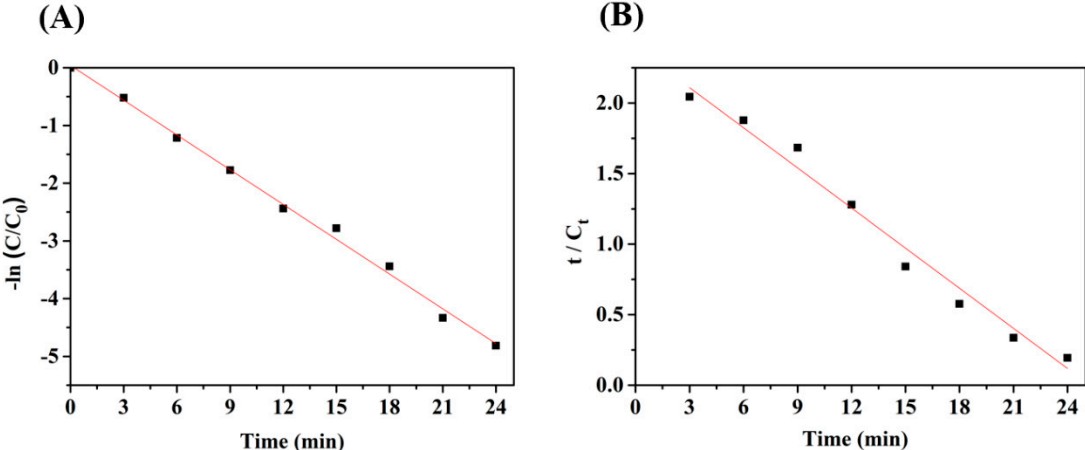

**Figure 5.** (**A**) Pseudo-first-order kinetic fitting and (**B**) pseudo-second-order kinetics fitting of azophloxine degradation. Reaction conditions: [PMS] = 5 mmol/L, T = 25 °C, pH = 6. Other chemical reagents used: [FeAlg-ACF] = 2 g/L, [azophloxine] = 50 μmol/L.

According to the azophloxine dye degradation conditions and the dynamic test data of the kinetics of primary and secondary fitting, it can be seen that for the FeAlg-ACFs/PMS system's oxidative degradation of azophloxine, the correlation coefficient of the first-order kinetics reaction is greater than the secondary dynamic correlation coefficient, showing that the dye degradation process is in line with a first-order kinetics reaction process. The degradation rate constant ($k_{obs}$) of azophloxine was found to be 0.19694.

Applying the principle of chemical kinetics to the field of the environment, not only can the mechanism of pollutant degradation be studied, but also the influencing factors of pollutant degradation can be determined and stabilization measures can be developed.

### 2.6. Mechanism of Catalytic Oxidation by the FeAlg-ACF/PMS System

The oxidation of organic pollutants was mainly contributed by ROS generated by PMS oxidation. In this study, we used t-butanol (TBA) and methanol (MA) to study the variation of ROS in the reaction system because MA is highly reactive with $SO_4^{\bullet-}$ and $\bullet OH$, while TBA only shows high reactivity with $\bullet OH$ [59].

As shown in Figure 6A, when 0.5 M TBA was added into our catalytic degradation system, the overall removal ratio of azophloxine dropped sharply from 100% to 42.3%. Therefore, $\bullet OH$ plays an important role in the photocatalytic oxidation reaction. Furthermore, when 0.5 M MA was added into the catalytic system, the removal rate declined to 26.8%. This suggests that the inhibition of azophloxine removal by MA was more significant than that by TBA because MA is highly reactive with both $\bullet OH$ and $SO_4^{\bullet-}$. Considering the possible influence of competitive adsorption, we further supplemented the free radical capture experiment under dark conditions, as shown in Figure 6B. In the figure, it can be observed that the overall effect under dark reaction conditions was worse than that under light, and the degradation rates with MA and TBA were further reduced, proving that both $\bullet OH$ and $SO_4^{\bullet-}$ are essential to the catalytic degradation of azophloxine. The reaction process of PMS activation is shown in Equations (12)–(15).

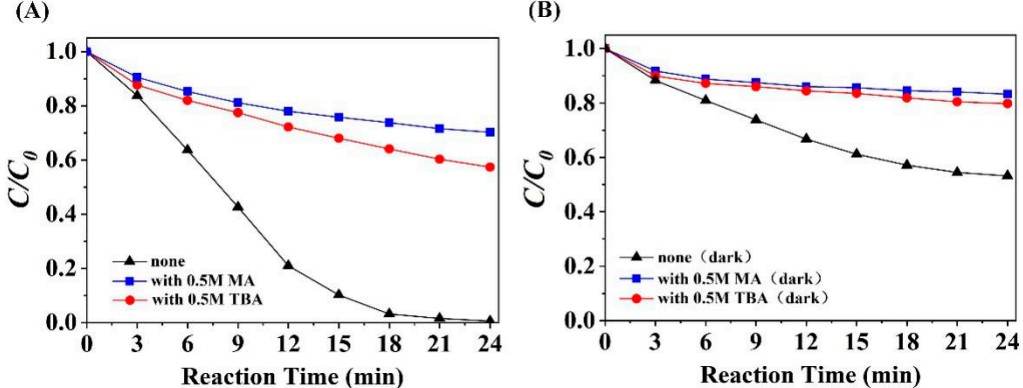

**Figure 6.** Effects of t-butanol (TBA) and methanol (MA) on azophloxine removal by the FeAlg-ACF/PMS system: (**A**) with light irradiation; (**B**) without light irradiation. Reaction conditions: [FeAlg-ACF] = 2 g/L, [PMS] = 5 mmol/L, [azophloxine] = 50 μmol/L, initial pH = 6.0, T = 25 °C.

$$M^{2+} + HSO_5^- \rightarrow M^{3+} + SO_4^{\bullet-} + OH^- \tag{12}$$

$$SO_4^{\bullet-} + OH^- \rightarrow SO_4^{2-} + \bullet OH \tag{13}$$

$$M^{3+} + HSO_5^- \rightarrow M^{2+} + SO_5^{\bullet-} + H_2O \tag{14}$$

$$SO_5^{\bullet-} + O_2^- \rightarrow SO_4^{\bullet-} + O_2 \tag{15}$$

To further understand the mechanism of ACF in the catalytic degradation system, we performed X-ray photoelectron spectroscopy analysis and determined the structural changes in FeAlg-ACF after the reaction. The peaks at 531.6 eV and 533.1 eV correspond to the oxygen atoms in the carbonyl group (C=O) and hydroxyl group (–O–H), respectively. As shown in Figure 7, the peak strength of oxygen in the C–O–H group decreased after the catalytic degradation of azophloxine. By contrast, the peak strength of oxygen proportionally increased in the C=O bond. This suggests that C–O–H groups were transformed into C=O during azophloxine removal. This change confirmed that PFRs in

ACF supplied electrons to Fe(III)-Alg, and electron transfer accelerated the Fe(III)-to-Fe(II) reduction process, which was the key step in accelerating the Fenton-like reaction.

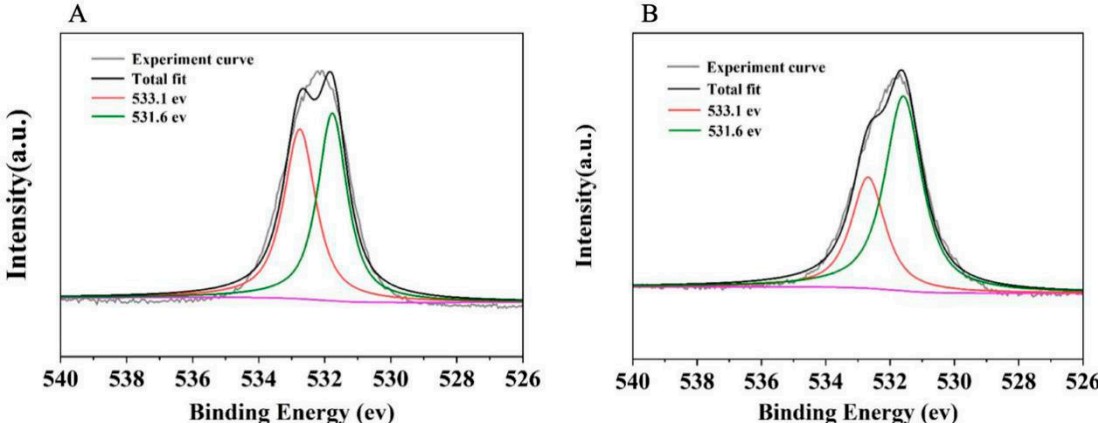

**Figure 7.** XPS spectra of O1s in FeAlg-ACF before (**A**) and after (**B**) its utilization in the FeAlg-ACF/PMS/visible light system. Reaction conditions: [FeAlg-ACF] = 2 g/L, [PMS] = 5 mmol/L, [azophloxine] = 50 μmol/L, initial pH = 6.0, T = 25 °C.

In conclusion, we propose a mechanism to explain azophloxine removal by the FeAlg-ACF/PMS system. First, the PFRs in ACF, as electronic storage, provide electrons to the Fe-Alg complex, which accelerates the LMCT process. At the same time, due to the high photosensitivity of the Fe-Alg complex, visible light radiation further enhances the LMCT process. Therefore, as a distinctive heterogeneous photocatalyst, FeAlg-ACF can effectively activate PMS to produce •OH and $SO_4^{\bullet-}$ to degrade organic dyes rapidly under visible light.

## 3. Experimental

### 3.1. Materials

Activated carbon fiber (ACF) was purchased from Jiangsu Sutong Carbon Fiber Co., Ltd. (Nantong, China). $FeCl_3 \cdot 6H_2O$ of analytical purity and sodium alginate $(C_6H_7NaO_6)_n$ of chemical purity were obtained from Sinopharm Chemical Reagent Co., Ltd. (Shanghai, China). Peroxymonosulfate (PMS) of analytical purity was provided by Aladdin biological technology Co., Ltd. (Shanghai, China). Azophloxine (analytical purity) was purchased from Macklin Biochemical Co., Ltd. (Shanghai, China). Rhodamine B (analytical purity), acid orange 7 (analytical purity), and methylene blue (analytical purity) were supplied by Sinopharm Chemical Reagent Co., Ltd. We used ultrapure water throughout the experiment.

### 3.2. Sample Preparation

The preparation scheme of the FeAlg-ACF catalyst is shown in Figure 8. A certain amount of ACF (10 g) was soaked in 3 mol/L nitric acid solution, then heated in a water bath at 30 °C for 12 h. The acidified ACF was repeatedly rinsed by ultrapure water and was then dried in a vacuum drying oven (Shanghai Yiheng Scientific Instrument Co., Ltd., Shanghai, China, DZF-6020) at 30 °C. Ferric chloride and sodium alginate were combined to form a suspension of ferric alginate at a 1:1 mass ratio via the ultrasonic stirring method. The acidified ACF was then immersed in the ferric alginate solution at 25 °C for 24 h. After rinsing by ultrapure water and drying at 30 °C, the novel heterogeneous catalyst FeAlg-ACF was obtained.

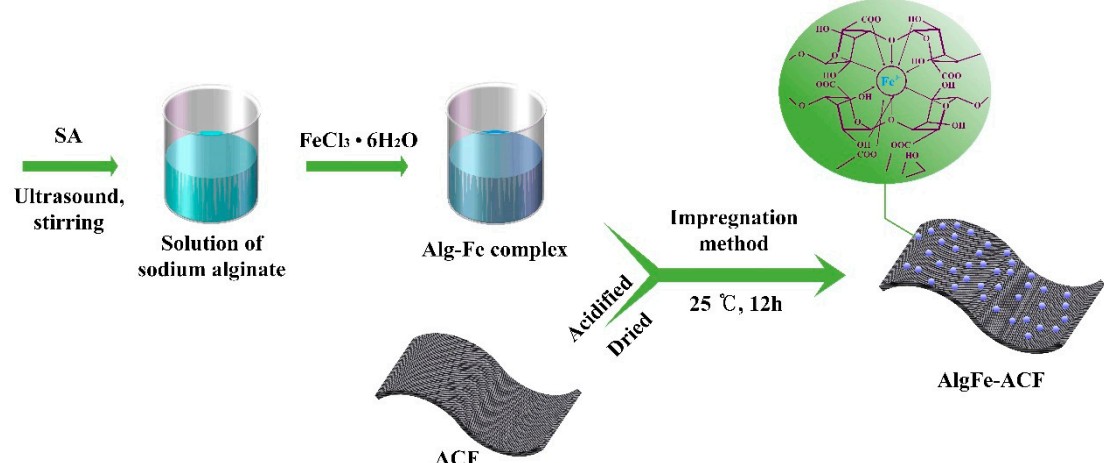

**Figure 8.** Schematic diagram of the preparation process of the heterogeneous photocatalysis FeAlg-ACF.

### 3.3. Dye Removal by FeAlg-ACF

The photocatalytic degradation of azophloxine was conducted in a photochemical reaction apparatus (Shanghai Yuncao, YCYN-GHX-D, Shanghai, China), equipped with a 500 W mercury lamp which irradiated the dyes at a distance of 10 cm through an ultraviolet glass cut-off filter (400 nm). The temperature was kept at 25 °C throughout the reaction. The reaction system was composed of azophloxine at an initial concentration of 50 μmol/L, FeAlg-ACF at a dosage of 2 g/L, and PMS at an initial content of 5 mmol/L. NaOH or HCl solution was used to adjust the initial pH value of the reaction system. The reaction system was treated in dark for 6 min, followed by photocatalytic degradation with a time interval of 3 min. The absorbance of azophloxine was determined using a UV–visible spectrophotometer (Shimadzu, Kyoto, Japan, UV-2550) at a maximum wavelength of 531 nm. According to the change in the absorption value of the sample before and after the reaction, the removal ratio (v) was calculated according to the absorbance of the solution before and after the treatment, using the following equation:

$$v = C_t/C_0 \times 100\% = A_t/A_0 \times 100\% \tag{16}$$

where $C_0$ is the initial concentration of azophloxine, $C_t$ is the azophloxine concentration after reaction time t, $A_0$ is used to show the initial absorbance of azophloxine, and $A_t$ is the absorbance of azophloxine after the reaction time [51–53].

### 3.4. Characterization

The surface morphology of the prepared catalyst was observed by scanning electron microscopy (ZEISS Sigma 500/VP Field emission scanning electron microscope). The iron content of the solution after the formation of FeAlg-ACF was determined by PANalytical X-ray fluorescence spectrometry (DY1040). The binding site of O1s was determined by X-ray photoelectron spectroscopy (XPS, Kratos Axis Ultra DLD, Kratos Analytical, Manchester, UK). The binding energy peaks of all XPS spectra were calibrated by using the binding energy peaks of 284.7 eV for principal C1s. The total organic carbon content of the dye solution was measured using a total organic carbon analyzer (Analytik Jena AG, Jena, Germany, model 3100) and the nitrate concentration was detected by ion chromatography (Thermo Fisher Scientific, Waltham, MA, USA, Aquion). The characteristic peaks of the reaction solution were explored by LC 3000 high-performance liquid chromatography (Beijing Tong Heng Innovation Technology Co., Ltd., Beijing, China, LC 3000).

## 4. Conclusions

In summary, we fabricated a novel heterogeneous catalyst, FeAlg-ACF, via a single-step process, i.e., the impregnation method. Using this catalyst together with PMS and visible light, we constructed a visible-light photocatalytic degradation system to remove azophloxine from aqueous solution. Azophloxine was completely removed after only 24 min processing by this novel system. In addition, the results indicated that more than 93.5% of the azophloxine was completely degraded into inorganic compounds; thus, potential secondary contamination by intermediate products during catalytic degradation can be inhibited. Compared to iron alginate gel beads, the dye removal efficiency was improved by the large specific surface area and PFRs provided by the carrier ACF. Moreover, the prepared heterogeneous catalyst FeAlg-ACF exhibits the advantages of reusability, a wide range of pH applications (2–10), and availability in visible light. With the additional advantage of recycled usage and ion applicability, the proposed FeAlg-ACF/PMS system can potentially be applied in the efficient and environmentally friendly removal of dye from wastewater.

**Author Contributions:** Conceptualization, D.W.; Data curation, D.W.; Formal analysis, D.W.; Investigation, K.X.; Methodology, K.X.; Software, C.F.; Supervision, Y.Y.; Writing—review & editing, X.C. All authors have read and agreed to the published version of the manuscript.

**Funding:** This research received no external funding.

**Acknowledgments:** The authors would like to thank the Ocean College Experimental Teaching Center of Zhejiang University for SEM characterization, and State Key Laboratory of Silicon Materials of Zhejiang University for XPS characterization. The authors also thank Lu et al. in the Department of Chemistry, Zhejiang University for helpful discussions and assistance in experiments. This research received no external funding.

**Conflicts of Interest:** We declare no conflict of interest exits in the submission of this manuscript, and manuscript is approved by all authors for publication.

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
