# Peer review of "Rapid Removal of Azophloxine via Catalytic Degradation by a Novel Heterogeneous Catalyst under Visible Light"

_catalysts, doi:10.3390/catal10010138_

Round 1
Reviewer 1 Report
This paper describes the synthesis of the material activated carbon fiber-supported ferric alginate and its application, together with peroxymonosulfate, as a photocatalyst in azophloxine degradation in aqueous solution. The paper has innovative information but, mainly from a quantitative point of view there is missing information:
The title should focus on azophloxine because it was the only dye that was analyzed. A kinetic model should be assayed because with need to obtained information about the kapp, the order of the reaction and other kinetic parameters. A linear kinetic model must be defined. The errors of the experimental parameters must be indicated. The results should be compared with a reference material in order to confirm the new material performance.
Author Response
Response: Thanks for your efforts in improving the quality of our manuscript.
Comment 1: The title should focus on azophloxine because it was the only dye that was analyzed.
Response 1: We would like to express our heartfelt thanks for this instructive concern. It is right that the title should focus on azophloxine. Therefore, in this work, the title has been changed to “Rapid removal of azophloxine via catalytic degradation by a novel heterogeneous catalyst under visible light”.
Comment 2: A kinetic model should be assayed because with need to obtained information about the kapp, the order of the reaction and other kinetic parameters. A linear kinetic model must be defined. The errors of the experimental parameters must be indicated.
Response 2: Thank you very much for this instructive concern. In our previous study, the degradation of target dye by FeCit-ACFs/PMS system and FeCit-ACFs/H2O2 system was in accordance with pseudo-first-order kinetics, and the reaction rate constants were 0.09555 min−1 and 0.00909 min−1 respectively, and degradation ratios were 100% and 99.82% respectively within 33 min. Based on this, we fitted the kinetics of degradation of azophloxine dyes by FeAlg-ACFs/PMS system, and compared the degradation efficiency and rate of the dyes.
Two models: pseudo-first-order, pseudo-second-order were used to investigate the degradation kinetics of azophloxine degradation by FeAlg-ACFs/PMS system. Pseudo-first-order model was first described by Lagergren (1898) and can be generally expressed as follows:
where qt is the amount of azophloxine removed at a time t (min) and k1 is the equilibrium rate constant (min-1). The values of log (qe – qt) are linearly with t, while k1 and qe can be determined from the slope and intercept of the plot of log (qe – qt) versus t, respectively.
Pseudo-second-order model, known as Ho’s pseudo-second-order model, was also applied to analyze the degradation kinetics of azophloxine by using the equation below:
where k2 is the rate constant of second order degradation (g mg-1 min-1 ) and k2q2e is the initial degradation rate constant (h). The plot of t/qt versus t should exhibit a linear relationship, qe and k2 can be determined from the slope and intercept of the plot, respectively.
These models were tested for the degradation of azophloxine and the best model was selected depending on the linear regression correlation coefficient, r2. Table 2 shows the parameters of these degradation kinetic models. The pseudo-first order kinetic model is: -ln (Ct/C0) = -0.19694t - 0.04125, R2 is 0.99524, while the pseudo-second-order kinetic model is: 1/Ct = -0.09478t - 21.84025, R2 is 0.98076. We can conclude that the pseudo-second-order model is not suitable for the degradation of azophloxine by FeAlg-ACFs/PMS system because of their relatively low r2 values. The experimental data fit well with the pseudo-first-order model with r2 > 0.99.
Fig. 5 Pseudo-first-order kinetic fitting and pseudo-secondary kinetics fitting of azophloxine degradation.
To better analyze the degradation rate of azophloxine, in-depth research on the theoretical study of the degradation of the target pollutant azophloxine by the reaction system was conducted. Pseudo-first-order dynamics fitting and pseudo-secondary dynamics fitting were performed on azophloxine within 24 min. The fitting results were presented in Fig. 5(line276), and the fitting kinetic constants were illustrated in Table 1(line 286).
|
dye |
Dynamic equation series |
The fitted reaction rate equation |
kobs |
Adj.R-Square |
|
azophloxine |
pseudo-first order kinetic model |
-ln(Ct/C0)=-0.19694t-0.04125 |
0.19694 |
0.99524 |
|
pseudo-second order kinetic model |
1/Ct =-0.09478t-21.84025 |
0.09478 |
0.98076 |
Table 1 kinetic constants of azophloxine degradation by FeAlg-ACFs/PMS/visible light system
According to azophloxine dye degradation condition, the dynamic test data of kinetics of primary and secondary fitting, it can be seen that FeAlg ACFs/PMS system oxidative degradation azophloxine holiday the first-order kinetics reaction of correlation coefficient is greater than the secondary dynamic correlation coefficient, show that the dye degradation process is in line with first-order kinetics reaction process. The degradation rate constant (kobs) of azophloxine was 0.19694.
Applying the principle of chemical kinetics to the field of environment, not only the mechanism of pollutant degradation can be studied, but also the influencing factors of pollutant degradation and stabilization measures can be developed.
The total experimental error is 2%, including the error of weighing the raw materials and reagents.
Comment 3: The results should be compared with a reference material in order to confirm the new material performance.
Response 3: We couldn’t be more grateful to you for this instructive comment. In this work, under the same carrier and oxidant, the results been compared with some reference material, such as ferric alginate (Fe-Alg), ferric citrate (Fe-Cit) and ferric oxalate (Fe-Msds), as shown in Fig. 2(line 155). The advantages have been summarized as follows:
For the self - structure and performance of ferric alginate
The advantages of ferric alginate over ferric citrate and ferric oxalate in degrading dyes are related to the structure of ferric alginate itself. In analogy to other Fe(III)-carboxylate complexes, such as ferric citrate and ferric oxalate, ferric alginate (Fe-Alg) could initiate the generation of a series of active oxygen species and Fe(II)/Fe(III) ions by the photo-induced ligand-metal electron transfer (LMCT) under light irradiation. However, ferric citrate and ferric oxalate are greatly affected by the pH value. When the pH value is greater than 4, both the ferric citrate and ferric oxalate complex will be converted into Fe2O3•nH2O amorphous precipitation, leading to the decrease or disappearance of the optical activity of the system.
As an iron-based photocatalyst, Fe-Alg complex is made by cross-linking the multivalent Fe(III) with natural, non-hazardous and edible polysaccharide alginate. There are a large number of carboxyl groups in the molecular structure of alginate. Studies have shown that some carboxyl acids can produce active reducing free radicals by self-photolysis, and there are also a large number of hydroxyl groups. Thanks to the unique gelling properties, alginates can react with metal ions to form stable organic-inorganic hybrid composite materials, which may be found promising applications in the environmental purification and remedy areas.
For the primary exploration comparison of Fe(III)-carboxylate complexes
In the introduction, combined with references, the significance of our topic selection is analyzed.
The effects of different systems on azophloxine removal were compared in Fig. 3B. Comparative experiments with ferric alginate (Fe-Alg), ferric citrate (Fe-Cit) and ferric oxalate (Fe-Msds) under the same carrier and oxidant were tested. The results indicated that the proposed catalytic oxidation system can remove 100% azophloxine within 24 min, while under the same system, the removal rates were only 92 % and 84 % when Fe-Alg was replaced with Fe-Cit and Fe-Msds respectively. It was related to the photosensitivity of Fe-Alg and its unique complexation structure. When ACF reacted with the oxidant PMS alone, the removal rate was 60% after 24 min. This phenomenon is attributed to the fact ACF is a porous carbon material, which can adsorb 8% azophloxine. On the other hand, ACF will continuously provide PFRs as electronic storage to participate in the LMCT process, under the irradiation of light, PMS is activated by electron transfer to produce small amounts of SO4•− and •OH to degrade dye.
Fig. 3. Decolorization of azophloxine by different Fe(III)-carboxylate systems as a function of time. Reaction conditions: [FeAlg-ACF] = 2 g/L, [PMS] = 5 mmol/L, [azophloxine] = 50 μmol/L, initial pH = 4.0, T = 25 ℃.
For the theoretical analysis between FeCit@ACFs/PMS/visible light and FeCit@ACFs/H2O2/visible light system
We made a comparison between FeCit@ACFs/PMS/visible light and FeCit@ACFs/H2O2/visible light system in our last article (Luo, Lianshun, Wu, Di, Dai, Dejun. Synergistic effects of persistent free radicals and visible radiation on peroxymonosulfate activation by ferric citrate for the decomposition of organic contaminants[J]. Applied Catalysis B Environmental, 205:404-411.) On the basis of the foregoing, we fabricated a novel heterogeneous catalyst-activated carbon fiber-supported ferric alginate (FeAlg-ACF). The Fe(III)-carboxylate complexes (ferric alginate (Fe-Alg), ferric citrate (Fe-Cit) and ferric oxalate (Fe-Msds)) were tested for degradation under the same conditions. The system which we studied before, FeCit@ACFs/PMS/visible light, can degrade azophloxine within 33 minutes, while our catalytic oxidation system, FeCit@ACFs/PMS/visible light, in the same conditions can degrade azophloxine within 20 minutes, and the reusability is stronger.
To further assess the difference in catalytic oxidation activity of these two systems, we adopted a general pseudo-first-order kinetic model (ln(Ct/C0)/kobst). As shown in Fig. 4 (line 249), the kobs for FeAlg-ACFs/PMS/visible light was calculated to be 0.19694 min−1, which was almost 2 fold higher than that for FeCit-ACFs/PMS/visible light (0.1098 min−1).The reaction kinetics equation of FeAlg-ACFs/PMS/visible light system was ln(Ct/C0) = -0.19694t + 0.02792, R2 is 0.99524,while the reaction kinetics equation of FeCit-ACFs/PMS/visible light system was ln(Ct/C0) = -0.1098t + 0.06193, R2 is 0.99531.
Fig. 4. Time profiles of azophloxine decolorization in FeAlg-ACFs/PMS/visible light and FeCit-ACFs/PMS/visible light; the inset shows the kobs of corresponding systems.
Table 2 kinetic constants of azophloxine degradation by two systems
|
  |
Intercept |
Slope |
Statistics |
||
|
  |
Value |
Standard Error |
Value |
Standard Error |
Adj. R-Square |
|
FeAlg-ACF |
0.02792 |
0.0487 |
-0.19694 |
0.00341 |
0.99524 |
|
FeCit-ACF |
0.06193 |
0.03806 |
-0.1098 |
0.00266 |
0.99531 |

Reviewer 2 Report
The comments are listed below:
The abstract should be rewritten. The reusability of the catalyst should be carried out. Mechanism need more descriptionAuthor Response
Reviewer #2: The abstract should be rewritten. The reusability of the catalyst should be carried out. Mechanism need more description.
Response: Thanks for your efforts in improving the quality of our manuscript.
Comment 1: The abstract should be rewritten. The reusability of the catalyst should be carried out. Mechanism need more description.
Response 1: The abstract has been modified. We have added the reusability of the catalyst (line24-25) and replenished the mechanism of our system (line 17-19) in the abstract. Thank you.

Reviewer 3 Report
This paper by Di Wu et al. constitutes a quite interesting research article about a novel heterogenous catalyst used for a removal of azoploxine. The paper seems to be rather useful and contains in an introduction section a high volume of various and sufficiently detailed information – it is demonstrated by a considerable list of references. I find the research designed properly. The paper is generally well written and the quality of the text is rather good.There are some grammatical and typographical errors that should be corrected. In my opinion the paper merits to be published in catalyst journal after minor changes. I have only two questions.
Line 56. I believe there should be photo-induced ligand-metal charge transfer (LMCT)" instead of electron transfer. Why in the whole manuscript Authors do not write the symbol of radical with a superscript in case of hydroxyl radicals, while in case of, for example, SO4.- the superscript is used?Author Response
Response: Thank you for your approval.
Comment 1: Line 56. I believe there should be photo-induced ligand-metal charge transfer (LMCT)" instead of electron transfer.
Response 1: The corresponding sentence has been modified. Thank you.
Comment 2: Why in the whole manuscript authors do not write the symbol of radical with a superscript in case of hydroxyl radicals, while in case of, for example, SO4.- the superscript is used?
Response 2: We have checked throughout the manuscript to ensure the radicals were written in the right form. Thank you.
Thank you again for your kind help on improving our article!

Round 2
Reviewer 1 Report
The authors have answered to the referee's questions.
Reviewer 2 Report
All of my conmments have been amended in the revised manuscript.